# Evaluating the Efficacy and Safety of Long-Acting GLP-1 Receptor Agonists in T1DM Patients

**Deene Mohandas [1],\*, Jamie Calma [2], Catherine Gao [1] and Marina Basina [1]**

[1] Division of Endocrinology, Gerontology, and Metabolism, Department of Medicine, Stanford University School of Medicine, Stanford, CA 94305, USA

[2] Division of Cardiovascular Medicine, Department of Medicine, Stanford University School of Medicine, Stanford, CA 94305, USA

\* Correspondence: deenem@stanford.edu

**Abstract:** (1) Background: Glucagon-like peptide 1 receptor agonists (GLP-1 RA) are a class of therapeutic agents that mimic the endogenous incretin hormone GLP-1. While this class of agents is not approved for Type 1 Diabetes (T1DM) due to concern of increased diabetic ketoacidosis (DKA) risk, long-acting GLP-1 medications are being commonly prescribed off label for T1DM in clinical practice. Several studies addressed the efficacy and safety of short-acting GLP-1 agonists therapy in patients with T1DM, but the data on long-acting agents are lacking. In our study, we aim to fill in this gap and help healthcare providers in their clinical decision making on the use of these agents for T1DM patients. (2) Methods: We conducted a retrospective chart review of T1DM patients on a long-acting GLP-1 for at least six months. Our retrospective chart review included information starting two years prior to starting GLP-1, and six or more months after starting GLP-1. Parameters collected included HbA1c, 14-day Continuous Glucose Monitor (CGM) and blood glucose (BG) data, and metabolic data (weight, systolic and diastolic blood pressure, and cholesterol levels). Statistical analysis was conducted using paired t-tests on R and Excel with $\alpha$ of 0.05. (3) Results: Our cohort consisted of 54 participants with T1DM on a long-acting GLP-1 (semaglutide, dulaglutide, exenatide extended-release [ER], albiglutide). Mean GLP-1 treatment duration was $23.85 \pm 15.46$ months. HbA1c values decreased significantly by an average of 0.71% percentage points (%-points, $p = 0.002$) comparing pre-therapy vs. on GLP-1 treatment. Similarly, for pre-therapy vs. on GLP-1 treatment values, CGM results were significant for increased time in range by 12.15%-points ($p = 0.0009$) showing a decreased average time in hyperglycemia (BG > 180 mg/dL) by a mean difference of 11.97%-points ($p = 0.006$), decreased 14-day mean BG by 19 mg/dl ($p = 0.01$), decreased 14-day BG standard deviation by 8.45 mg/dl ($p = 0.01$), decreased incidence of DKA hospitalization, and a decrease in weight by 3.16 kg ($p = 0.007$). (4) Conclusions: As more data emerges on cardiovascular and renal benefits of long acting GLP-1 in type 2 diabetes, there have been no reported outcomes in T1DM. Our study is the first to demonstrate glycemic and metabolic benefits of this class of medication as an adjunct therapy to insulin in T1DM, and safety of its use over an average of 1.5–2 years' time. This study represents real life experience and the data warrants confirmation by additional prospective studies.

**Keywords:** GLP-1; type 1 diabetes; adjunct therapy to insulin; semaglutide

## 1. Introduction

Glucagon-like peptide 1 receptor agonists (GLP-1 RA) are a class of therapeutic agents that function by mimicking the endogenous incretin hormone GLP-1, which increases insulin secretion, inhibits glucagon release, and delays gastric emptying in response to oral ingestion of food [1,2].

Both long- and short-acting GLP-1 RAs are used in treatment of type 2 diabetes mellitus (T2DM) [1]. Among T2DM patients, long-acting GLP-1 medications (semaglutide, dulaglutide, exenatide extended-release [ER], albiglutide) have been shown to reduce

HbA1c, improve glycemic control, and reduce body weight, with limited side effects (low rates of minor hypoglycemia events, transient nausea and vomiting). The Semaglutide Unabated Sustainability in Treatment of Type 2 Diabetes (SUSTAIN) clinical trials tested the safety profile of once-weekly semaglutide injections in individuals with T2DM. The SUSTAIN trials consisted of seven randomized controlled phase 3 trials with over 8000 T2DM patients. The longest of these trials, SUSTAIN 6, followed patients on semaglutide for 104 weeks. In the SUSTAIN trials, semaglutide was consistently shown to improve glycemic control, decrease cardiovascular risk, and promote weight loss at standard doses [3]. Analysis across SUSTAIN 1–5 and 7 was significant for reduction in HbA1c (1.2–1.8%) and weight loss (3.5–6.5 kg). The efficacy and safety of long-acting GLP-1 dulaglutide comparing to short-acting liraglutide as add-on to metformin was shown in an 18-month retrospective observational cohort study [4]. A significantly larger proportion of patients in the dulaglutide group reached the glycemic target of <7% comparing to patients in the liraglutide group [4]. A 2022 review of multiple studies investigating the cardiovascular impact of long-acting GLP-1 RAs in T2DM demonstrated that injectable semaglutide, dulaglutide, and liraglutide reduce the risk of a major adverse cardiovascular event [5]. These results suggest the efficacy of long-acting GLP-1 medications in improving glycemic control as well as controlling risk factors for cardiovascular and other diabetes complications (body weight, blood pressure).

GLP-1 RAs have been studied in type 1 diabetes mellitus (T1DM) patients to a lesser extent. GLP-1 RAs have potential as an adjunctive therapy for T1DM patients to insulin therapy. Although insulin therapy is standard in management of T1DM and glycemic control clearly shows reduction in the rate of microvascular complications, insulin use is also associated with increased risk of hypoglycemia and weight gain [6]. Weight gain is a known risk factor for cardiovascular disease (CVD) [7], and cardiovascular disease is an established cause of premature death among T1DM patients—causing 30–44% of all deaths among T1DM individuals [8]. Therefore, these medications that can improve glycemic control while reducing insulin requirements and weight may be promising adjunct treatments in T1DM [9,10]. It is thought that GLP-1 RAs induce weight loss via a satiety effect and a 26-week study performed with overweight/obese T1DM adults showed increased lipid oxidation and thermogenesis as the main mechanism for weight loss with liraglutide [11]. A separate 24-week study of T1DM adults on liraglutide showed benefits of prolonged satiety and decreased desire and consumption of food [12]. These findings suggest that the GLP-1 RA medication class may be a promising adjunct therapy to insulin for decreasing the high burden of CVD among T1DM patients.

Several studies have focused on short-acting GLP-1 RAs in T1DM patients. One narrative review conducted by Guyton and Brooks in 2019 found that in T1DM patients, short-acting GLP-1 agonists, exenatide and liraglutide, were associated with weight loss, decrease in total daily insulin requirements, and moderate improvements in glycemic control [13]. The largest trial of short-acting GLP-1 medications in T1DM was ADJUNCT ONE, which followed 1400 T1DM patients randomly assigned to either placebo use or liraglutide 1.8, 1.2, or 0.6 mg in addition to insulin therapy. The study found that while liraglutide reduced body weight, daily insulin dose, and HbA1c, the medication also increased the rate of hypoglycemia, as well as DKA on the 1.8 mg dose [14]. A second trial called ADJUNCT TWO, which followed 835 T1DM patients, reported similar results [15]. Due to these findings of increased hypoglycemia and diabetic ketoacidosis (DKA) risk associated with short-acting GLP-1 medications, GLP-1 medications are not currently FDA approved for use in T1DM patients.

To our knowledge, only a handful of studies have assessed the impact of a long-acting GLP-1 RA in T1DM patients. In a retrospective observational study with eleven patients, Traina et al. found that among eleven T1DM patients with residual beta cell activity, patients who had taken exenatide ER for three months saw reductions in HbA1c levels, body weight, BMI, and total daily insulin dose. However, four participants discontinued the medication early due to injection site nodule formation, and one participant discontinued early due to

gastrointestinal intolerance. The study did not report on DKA or hypoglycemia episodes as outcomes [16]. Another study by Herold et al. examined the effects of exenatide ER on glucose control compared to placebo in 79 T1DM patients in a randomized double-blind phase 2b study. In this study, participants treated with exenatide ER for 24 weeks, and then followed for an additional 24 weeks off the study drug. The study found a significant HbA1c reduction (7.71% vs. 8.05%) after 12 weeks on exenatide ER compared to the placebo group. Of note, while the HbA1c reduction in the exenatide ER was greater in participants with a detectable C-peptide level, this difference was not significant. In all participants (detectable and undetectable C-peptide), there was a < 0.1% decrease in HbA1c compared to baseline at week 52, 24 weeks after stopping exenatide ER. This study did not report on DKA, or metabolic outcomes, and also did not follow patients who were on exenatide ER for longer than 24 weeks. Although the study reported that adverse gastrointestinal effects were more frequent for those on exenatide ER, there was no increase in the rate of hypoglycemia [17]. A review article on the impact of GLP-1 RAs in T1DM poses the need to study newer generations of long-acting GLP-1 medications (including dulaglutide, semaglutide, and albiglutide) [10].

There are even more limited data regarding the use of long-acting GLP-1 agonists in patients with T1DM who also have insulin resistance component, often termed as having "double diabetes". One cross-sectional study of 107 adults with T1DM demonstrated that overweight/obese T1DM individuals had higher total daily insulin doses and a higher prevalence of "double diabetes". Additionally, the same study concluded that "double diabetes" is associated with increased cardiovascular risk profile and is common even in non-overweight/non-obese individuals with T1DM [18]. While studies on long-acting GLP-1s and insulin resistance in T1DM are limited, an ongoing phase 3 randomized open-label study sponsored by Centre Hospitalier Universitaire Dijon is investigating the use of semaglutide in reducing insulin resistance and improving glycemic control in patients with "double diabetes" [19].

Our retrospective chart review study aims to evaluate glycemic and metabolic benefits of long-acting GLP-1 therapy in a cohort of 54 T1DM patients and assess safety concerns. Unlike prior studies, our study examines other newer long-acting GLP-1 medications besides exenatide ER. Our study also looks at a wider range of metrics for glycemic control (including time in range, hyperglycemia, and hypoglycemia) as well as cardiovascular risk factors such as weight, cholesterol levels, and blood pressure. Finally, our study includes patients taking long-acting GLP-1 medications for longer than 24 weeks and up to 2 years.

## 2. Materials and Methods

### 2.1. Study Design

We conducted a retrospective chart review using Stanford Medicine's Data Repository (STARR), a cohort discovery tool, to identify adult T1DM patients seen at Stanford Health Care's Department of Endocrinology. Inclusion criteria were patient age ≥ 18 years, diagnosis of T1DM, and use of a long-acting GLP-1 medication for ≥ 6 months. T1DM diagnosis was confirmed by the presence of ICD-10 code E10 (Type 1 diabetes mellitus) in the patient chart. A few patients did not have ICD-10 code E10 in their chart, but were included in the study if a diagnosis of T1DM was in the visit notes, and laboratory testing showed C-peptide level <0.2 nmol/L or presence of positive antibodies for T1DM. Exclusion criteria were pregnancy, history of concurrent steroid use, and age less than 18 years. Data collection was done via manual chart review using the institution's Electronic Health Record system (EPIC). Study data were securely collected and managed using REDCap electronic data capture tools hosted at Stanford University [20–22]. Parameters collected included HbA1c, 14-day CGM and BG data at each visit, and metabolic data such as weight, systolic and diastolic blood pressure, and cholesterol levels. Data from two years prior to starting GLP-1, and six or more months after starting GLP-1 were included. Approval from the local Institutional Review Board was obtained prior to initiation of this study.

### 2.2. Statistical Analysis

In regard to data analysis, descriptive and inferential statistics were utilized to analyze the data and understand their significance. The average of 2-year data prior to the initiation of GLP-1 was compared with data from an average duration of $23.85 \pm 15.46$ months on long-acting GLP-1 therapy. At the time of review, one patient had been on various forms of long-acting GLP-1 therapy for over 7 years. Statistical analysis was conducted through paired t-tests using R and Excel to compare baseline values with values on long acting GLP-1.

### 3. Results

We identified 54 participants with T1DM who had taken a long-acting GLP-1 RA such as once-weekly semaglutide (Ozempic), dulaglutide (Trulicity), exenatide ER (Bydreon), or albiglutide (Tanzeum) for at least six months. The mean GLP-1 duration was $23.85 \pm 15.46$ months. Participants had a mean age of 41.5 years and mean time since T1DM diagnosis of 16.4 years. A total of 64.8% ($n = 35$) were female and 57.4% ($n = 31$) were white, 20.3% ($n = 11$) self-identified as Black or Hispanic/Latino per chart review. At time of GLP-1 RA initiation, 29.6% ($n = 16$) of participants were using multiple daily injections (MDIs), 66.7% ($n = 36$) were on an insulin pump, and 77.8% ($n = 42$) were using a CGM device. A majority, 51.6% (28 patients), were using an insulin pump and CGM as a closed-loop system. Fewer patients, 33.3% ($n = 18$), used a CGM device plus MDIs. Only 66.7% of participants ($n = 36$) had recorded C-peptide levels. The median C-peptide was 0.073 nmol/L, calculated from most recent available C-peptide value. Of those with an available C-peptide value, 63.9% ($n = 23$) were C-peptide negative (<0.2 nmol/L) (Table 1).

**Table 1.** Participant Demographics.

| Demographics ($n$ = 54) | Mean $\pm$ SD |
| --- | --- |
| Age—years | $41.54 \pm 13.89$ |
| Time since Type 1 diabetes mellitus (T1DM) Diagnosis—years ($n$ = 18) | $16.37 \pm 12.92$ |
| Time on long-acting glucagon-like peptide 1 receptor agonist (GLP-1)—months | $23.85 \pm 15.46$ |
| Sex—$n$ (%) [1] | |
| Female | 35 (64.8%) |
| Male | 19 (35.2%) |
| Race—$n$ (%) [1] | |
| White | 31 (58.5%) |
| Asian | 4 (7.5%) |
| Black | 1 (1.9%) |
| Other | 14 (26.4%) |
| Unknown | 3 (5.7%) |
| Ethnicity—$n$ (%) [1] | |
| Hispanic/Latino | 10 (18.9%) |
| Non-Hispanic | 39 (73.6%) |
| Unknown | 4 (7.5%) |
| Insulin Treatment Type—$n$ (%) [2] | |
| Multiple Daily Injections (MDIs) | 16 (29.6%) |
| Insulin Pump | 36 (66.7%) |
| Closed loop | 28 (51.6%) |
| Continuous Glucose Monitor (CGM) | 42 (77.8%) |
| Glucometer | 6 (11.1%) |
| GLP-1 Type—$n$ (%) | |
| Trulicity | 19 (35.2%) |
| Bydreon | 2 (3.7%) |
| Ozempic | 34 (63.0%) |
| Tanzeum | 2 (3.7%) |
| Time on GLP-1—months | $23.85 \pm 15.46$ |
| C-peptide level ($n$ = 36) (mean $\pm$ SD) | 0.32 nm/L $\pm$ 0.51 |
| Positive C-peptide | |
| Yes | 13 (36.1%) |
| No | 23 (63.9%) |

[1] Categories for sex, race, and ethnicity are based on the demographic categories available in EPIC, and may not reflect how participants self-identify. [2] Some participants were using multiple insulin treatment types. Percentages may add to over 100%.

The long-acting GLP-1 medications taken by participants were Ozempic (63.0%), Trulicity (35.2%), Bydreon (3.7%), and Tanzeum (3.7%) (Table 1). Some participants switched

between medication types or experienced dosage adjustments during the time frame of interest. Furthermore, 27.8% of participants discontinued their long-acting GLP-1 medication over a two-year period. The most common reasons for discontinuing were gastrointestinal symptoms including nausea and vomiting (40.0%) and minimal or negative impact on glycemic control (20.0%). An additional 33.3% discontinued for unknown reasons. A less frequently documented reason for discontinuation was lack of insurance coverage (6.7%) (Table 2).

**Table 2.** Reasons for discontinuing GLP-1.

| (*n* = 54) | *n* (%) |
| --- | --- |
| Discontinued GLP-1 | |
| Yes | 15 (27.8%) |
| No | 39 (72.2%) |
| Reason for Discontinuing | |
| GI Symptoms | 35 (64.8%) |
| Minimal or negative impact on glycemic control | 19 (35.2%) |
| Minimal or negative impact on weight control | 0 (0.0%) |
| Lack of insurance coverage | 1 (6.7%) |
| Unknown | 5 (33.3%) |

Baseline and post GLP-1 medication values were calculated for each parameter by averaging values over a two-year period before starting the medication, and values collected starting six months after initiation of long-acting GLP-1 therapy and ending at the time of chart review. Using this approach, we observed that HbA1c values decreased significantly from a baseline of 7.76 ± 1.40% to 7.05 ± 1.00% (mean difference = −0.71% percentage-points [%-points], $p$ = 0.002, $n$ = 43) (Table 3).

**Table 3.** Participant HbA1c, glycemic trends, insulin dose, cholesterol and creatinine levels, weight, and blood pressure at baseline and after 6 months of long-acting GLP-1.

| Parameter (*n* of Patients with Available Data) | Baseline (Mean ± SD) | Post GLP-1 (Mean ± SD) | Mean Difference | *p*-Value |
| --- | --- | --- | --- | --- |
| HbA1c (%-points) (*n* = 43) | 7.76 ± 1.40 | 7.05 ± 1.00 | −0.71 | 0.0018 |
| Time in Range (TIR) (%) (*n* = 23) | 54.59 ± 23.12 | 66.74 ± 17.82 | +12.15 | 0.000850 |
| Time in Hypoglycemia (%) (*n* = 28) | 2.32 ± 2.63 | 1.78 ± 2.15 | −0.54 | 0.0900 |
| Time in Hyperglycemia (%) (*n* = 20) | 42.20 ± 21.47 | 30.23 ± 15.66 | −11.97 | 0.00628 |
| 14-day Avg Blood Glucose (BG) (mg/dL) (*n* = 27) | 182 ± 32.0 | 163 ± 24.9 | −19 | 0.0145 |
| CGM SD (*n* = 17) | 56.09 ± 18.26 | 47.64 ± 12.62 | −8.45 | 0.006665 |
| Total Daily Insulin Dose (TDD) (units) (*n* = 22) | 58.18 ± 32.32 | 55.17 ± 29.92 | −3.01 | 0.2963 |
| Insulin Requirement (units/kg) (*n* = 18) | 0.553 ± 0.22 | 0.547 ± 0.19 | 0.006 | 0.827 |
| Creatinine (*n* = 37) | 0.97 ± 0.68 | 1.09 ± 1.47 | +0.12 | 0.54 |
| Total Cholesterol (*n* = 27) | 157.85 ± 32.97 | 155.07 ± 31.70 | −2.78 | 0.63 |
| LDL-Cholesterol (*n* = 24) | 81.71 ± 26.70 | 85.83 ± 29.50 | +4.12 | 0.38 |
| Weight (kgs) (*n* = 36) | 86.66 ± 19.24 | 83.50 ± 20.83 | −3.16 | 0.007 |
| Systolic Blood Pressure (mmHg) (*n* = 19) | 126.05 ± 17.24 | 126.79 ± 15.48 | +0.74 | 0.761 |
| Diastolic Blood Pressure (mmHg) (*n* = 19) | 72.74 ± 12.17 | 70.74 ± 12.48 | −2.00 | 0.408 |

Average time in range (TIR) increased significantly from a baseline of 54.59% ±23.12 to 66.74% ±17.82 (mean difference = +12.15%-points, $p$ = 0.0009, $n$ = 23). Average time in hyperglycemia decreased significantly from a baseline of 42.20% ± 21.47 to 30.23% ± 15.66 (mean difference = −11.97%-points, $p$ = 0.006, $n$ = 20). The 14-day BG average decreased from a baseline of 182 ± 32.00 mg/dl to 163 ± 24.9 mg/dl (mean difference = −19 mg/dl, $p$ = 0.015, $n$ = 27). There was a non-significant decrease in insulin requirement from 0.553 ± 0.22 units/kg to 0.547 ± 0.19 unit/kg (mean difference = −0.0061, $p$ = 0.83, $n$ = 18). There were no statistically significant differences in time in hypoglycemia (BG < 70) for baseline and post-GLP-1 CGM values ($p$ = 0.09, $n$ = 28) (Table 3). Per CGM data, there were also non-significant decreases in time spent in very low, low, and very high ranges

(Figure 1). Although these changes were small, they represent an overall trend of improved glycemic control.

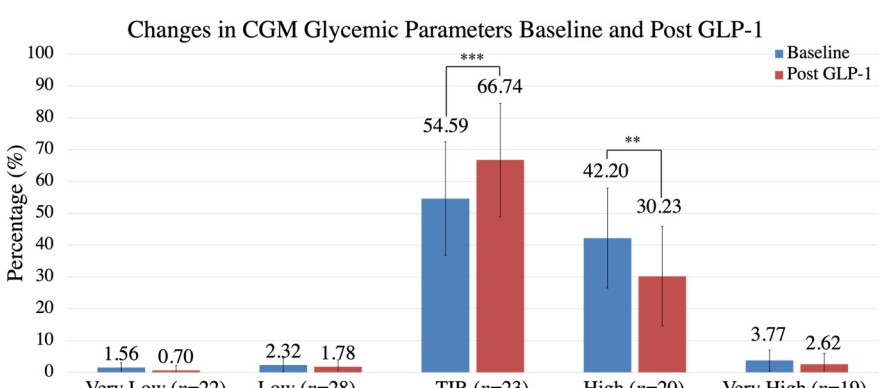

**Figure 1.** Changes in CGM glycemic parameters at baseline and post GLP-1. ** $p < 0.01$, *** $p < 0.001$.

Average weight decreased significantly from a baseline of 86.66 ± 19.24 kg to 83.50 ± 20.83 kg, (mean difference = −3.16 kg, $p = 0.007$, $n = 36$). There were no statistically significant differences between baseline and post-GLP-1 CGM values for serum creatinine ($p = 0.54$, $n = 37$), total cholesterol ($p = 0.63$, $n = 27$), LDL-Cholesterol ($p = 0.38$, $n = 24$), systolic blood pressure ($p = 0.76$, $n = 19$), or diastolic blood pressure ($p = 0.41$, $n = 19$) (Table 3).

## 4. Discussion

Our study represents a real-life experience and is the first to demonstrate both glycemic and metabolic benefits of the GLP-1 RA class of medications in T1DM patients, as well as safety of its use over an average duration of 1.5–2 years. Furthermore, there was no difference in hypoglycemia risk after patients started on a long-acting GLP-1 medication. There was also no increase in DKA risk, even with many participants being C-peptide negative. While the results of our study addressed the safety concerns about the use of GLP-1 in T1DM and demonstrated no increased risk of hypoglycemia or incidence of DKA while on long-acting GLP-1 therapy and showed significant benefits, the data are currently not sufficient to draw conclusions on cardiovascular risk outcomes.

In our study cohort we observed a significant decrease in HbA1c (−0.71%-points, $p = 0.002$) as well as a significant decrease in average weight (−3.16 kg, $p = 0.0068$). In comparison to the ADJUNCT ONE and ADJUNCT TWO trials done with once-daily liraglutide in a large population of T1DM patients, decrease in HbA1c and weight were also observed. In ADJUNCT ONE, the greatest reduction in HbA1c was 0.3%-points after 26 weeks on liraglutide 1.8 mg. However, by week 52 on liraglutide 1.8 mg, this reduction in HbA1c had diminished to 0.2%-points. In the 26-week long ADJUNCT TWO trial, HbA1c decreased by 0.35%-points on liraglutide 1.8 mg [15]. Similarly, both ADJUNCT ONE and ADJUNCT TWO noted greatest reduction in weight, 5.0 and 4.8 kg, respectively, after 26 weeks on liraglutide 1.8 mg [22]. In addition to HbA1c and weight, both ADJUNCT trials also reported on insulin requirement. At 26 weeks on liraglutide 1.8 mg, total daily insulin dose decreased by 10% in ADJUNCT ONE and 12% in ADJUNCT TWO [22]. This reduction in total daily dose was primarily driven by decreased requirement for bolus insulin [23].

With regard to HbA1c, our data set demonstrated a larger decrease in HbA1c over a greater amount of time while on a long-acting GLP-1 as compared to both ADJUNCT ONE and ADJUNCT TWO. Additionally, although reduction in weight was a significant finding in our cohort (−3.16 kg, $p = 0.007$), larger reductions in weight were observed in the ADJUNCT trials. Unlike the ADJUNCT trials, review of our limited available metabolic data showed a non-significant decrease in total daily insulin requirement of 2.65%.

While retrospective chart reviews provide great insight in patients on GLP-1 medication, there are several limitations to the study especially with our cohort data being confounded by COVID-19 and the pandemic. Telemedicine has helped a great deal in

allowing access to healthcare during the pandemic. Unfortunately, due to the nature of virtual appointments, our data became limited to patient-reported weight and blood pressure, which introduced unstandardized and inconsistent measurements in each participant. In general, metabolic data were significantly limited in the setting of increased virtual and telemedicine visits during the COVID-19 pandemic and metabolic outcomes should be further evaluated in a larger cohort. Furthermore, the study's retrospective design did not allow for a controlled setting to ensure consistent standardization of data collection and medication adherence of the patient cohort. The relatively short duration of our study data, with an average of approximately 2 years on a long-acting GLP-1 RA, is a significant limitation that does not allow for the extrapolation of longer-term cardiovascular risk or benefit associated with the therapy. Lastly, the small number of patients reviewed in this study can also be considered a potential limitation of the study. Although we observed no adverse side effects in the patient cohort being on GLP-1 for an average of almost 2 years, this cannot represent a longer timeline and will warrant a longer duration study with a larger patient cohort.

In summary, this retrospective chart review of 54 patients with T1DM on a long-acting GLP-1 demonstrated significant benefits including decreased HbA1c, increased TIR, decreased time in hyperglycemia, decreased average 14-day BG, and decrease in weight, with no significant change in incidence of DKA or hypoglycemia. Additionally, while symptoms such as nausea and vomiting or lack of improvement in glycemic control did lead to discontinuation of therapy for some participants, the majority of participants, over 72%, remained on their long-acting GLP-1 medication over a two-year period. The significant finding of decrease in weight while on a long-acting GLP-1 and the association of weight gain with increased risk of CVD, has potential implications for reduction of CVD risk. Our findings of improved glycemic control measured by reduction in HbA1c, increased TIR, and decreased time in hyperglycemia, along with reduction in weight, without increased risk of DKA, support the use of long-acting GLP-1 RA medications as an adjunct therapy to insulin. Currently, GLP-1 medications are not approved for T1DM based on the results of trials with short-acting GLP-1 RAs. Our study suggests that long-acting GLP-1s may have differing effects on hypoglycemia and DKA risk. Further studies may benefit from stratification of C-peptide levels as a potential component in assessing risk. Although some long-acting GLP-1 medications are used off-label in T1DM patients, these medications should be considered for approval in this patient population after obtaining additional safety data in randomized controlled trials, as an adjunct therapy to insulin, due to their glycemic and potential metabolic benefits.

## 5. Conclusions

There is an urgent need for strategies to improve glycemic control and reduce morbidity and mortality in T1DM without increasing the risk for hypoglycemia. Long-acting GLP-1 RAs are a promising adjunct therapy to insulin, as they may improve glycemic control and limit weight gain associated with insulin therapy, and promote weight loss [14,15]. These effects may decrease the elevated risk of CVD in T1DM patients. As more data emerges on cardiovascular and renal benefits of long acting GLP-1 in T2DM patients, there have been few reported outcomes in T1DM patients.

**Author Contributions:** Conceptualization and development of aims and research question was achieved by D.M., C.G. and M.B. J.C. helped with project administration, and facilitated the methodology and how the study should be executed. The formal analysis, investigation, data curation, and visualization were equally shared by D.M., C.G and J.C. Writing the original manuscript and draft preparation was done by D.M., C.G. and J.C. Reviews and edits were done by D.M., with support from C.G., J.C. and M.B. Finally, M.B. provided supervision to the study team. All authors have read and agreed to the published version of the manuscript.

**Funding:** This research received no external funding.

**Institutional Review Board Statement:** The study was conducted in accordance with the Declaration of Helsinki, and approved by the Institutional Review Board (or Ethics Committee) of Stanford University (protocol ID 62987 with a date of approval of 29 November 2021).

**Informed Consent Statement:** Patient consent was waived due to the study being a retrospective chart review and poses no risk to the participants in the study as all data were de-identified. The study did not have any effect on the participants' clinical care.

**Data Availability Statement:** The data presented in this study are available on request from the corresponding author. The data are being securely kept in REDCap and are not publicly available to protect any identifiers from improper use and disclosure.

**Acknowledgments:** The authors would like to thank Stanford research information technology.

**Conflicts of Interest:** Marina Basina is an editor of the special issue on Type 1 Diabetes.

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
