# Peer review of "Evaluating the Efficacy and Safety of Long-Acting GLP-1 Receptor Agonists in T1DM Patients"

_endocrines, doi:10.3390/endocrines4010008_

Round 1

Reviewer 1 Report

This paper is interesting and provides important information. My suggestions are the following: 

Introduction: Background is well presented. Line 60: It would be useful to the reader to include also information about efficacy and safety in GLP1-RA add-on studies  in type 2 DM (example: Mirabelli M. et al, J Clin Med 2021).

Results: Table 1: C-peptide level (median should be mean); Table 3: SD (N=17) is referred to CGM, please specify. LDL should be LDL-Cholesterol (also in the text, line 225). Figures 1 and 2 do not add clarity to results and should be omitted. 

Bibliography: Reference list should be fixed according to the journals' instructions. Instead, as example, in reference n. 4 name initials precede the surname. Also, in reference n. 24, names are displayed instead of surnames.

Reviewer 2 Report

In this article "Evaluating the Efficacy and Safety of Long-Acting GLP-1 Receptor Agonists in T1DM Patients", Mohandas et al., conducted a retrospective, observational study on T1DM patients. They investigate the efficacy and safety of long-acting GLP-1 agonists therapy in patients with T1D and found that glycemic and metabolic benefits of this class of medication as an adjunct therapy to insulin in T1D, and safety of its use over an average of 1.5-2 years’ time. However, some concerns have to be addressed.

Specific comments:

Abstract:

I suggest that authors should cut down the word numbers, for example, the exact quantitative value (like 7.76 ± 1.48 vs 7.05 ± 1.00%-points, mean difference = -0.71%-points, p=0.002, N=43) can be omitted. Moreover, the abbreviation of type 1 diabetes is T1D or T1DM? You should unify the usage in your title, abstract and main text.

Instruction:

In Line 107, 109 and others, what is the meaning of ER? When firstly showed, you should give a full name. Also, I suggest authors to simplify your words.

Methods:

In study design part, are the ages of all participants above 18? I suggest authors to clarify inclusion and exclusion criteria in detail. And I think the introduction of REDCap is useless.

Discussion:

Did authors discuss the limitations of this study?

Round 2

Reviewer 2 Report

Current form is suitable for publication.